# Plasticity of Adipose Tissues: Interconversion among White, Brown, and Beige Fat and Its Role in Energy Homeostasis

**DOI:** 10.3390/biom14040483

**Published:** 2024-04-16

**Authors:** Yanqiu Peng, Lixia Zhao, Min Li, Yunfei Liu, Yuke Shi, Jian Zhang

**Affiliations:** School of Bioengineering, Zunyi Medical University, Zhuhai 519000, China; pengyanqiu@zmuzh.edu.cn (Y.P.); zhaolixia@zmuzh.edu.cn (L.Z.); limin2@zmuzh.edu.cn (M.L.); liuyunfei@zmuzh.edu.cn (Y.L.); shiyuke@zmuzh.edu.cn (Y.S.)

**Keywords:** obesity, WAT, BAT, beige adipose tissue, interconversion

## Abstract

Obesity, characterized by the excessive accumulation of adipose tissue, has emerged as a major public health concern worldwide. To develop effective strategies for treating obesity, it is essential to comprehend the biological properties of different adipose tissue types and their respective roles in maintaining energy balance. Adipose tissue serves as a crucial organ for energy storage and metabolism in the human body, with functions extending beyond simple fat storage to encompass the regulation of energy homeostasis and the secretion of endocrine factors. This review provides an overview of the key characteristics, functional differences, and interconversion processes among white adipose tissue (WAT), brown adipose tissue (BAT), and beige adipose tissue. Moreover, it delves into the molecular mechanisms and recent research advancements concerning the browning of WAT, activation of BAT, and whitening of BAT. Although targeting adipose tissue metabolism holds promise as a potential approach for obesity treatment, further investigations are necessary to unravel the intricate biological features of various adipose tissue types and elucidate the molecular pathways governing their interconversion. Such research endeavors will pave the way for the development of more efficient and targeted therapeutic interventions in the fight against obesity.

## 1. Introduction

Obesity is a condition characterized by excessive fat accumulation that arises when energy intake consistently surpasses energy expenditure [1]. Although the symptoms of obesity may not be immediately apparent, its long-term consequences are severe. As a key player in various physiological processes, recent studies have reported that obesity-related metabolic complications are intertwined with multiple mechanisms, including epigenetics [2,3,4,5,6,7,8], immune regulation [9,10,11,12,13], gut microbiota metabolism [14,15,16], aging, and cancer [17,18,19,20,21]. While not all obese individuals develop severe complications, those who are metabolically obese are more prone to metabolic disorders, whereas metabolically healthy obese individuals rarely experience such issues [22]. This may be related to the accumulation of different types of adipose tissue (WAT or BAT) in different regions of the body (visceral adipose tissue (VAT) or subcutaneous adipose tissue (SAT)) and the associated metabolic changes they induce. Therefore, effective obesity treatment is a public health priority, with a focus on fundamental research into adipose tissue biology and plasticity.

Various methods for preventing obesity have been developed, with lifestyle changes, including modifications to dietary habits and physical exercise [23,24], being the most common starting points for weight loss. However, the success rate is relatively low, possibly due to poor long-term adherence [25]. Pharmacotherapy is a major treatment intervention; for example, glucagon-like peptide-1 (GLP-1) receptor agonists such as liraglutide and semaglutide have been approved for the treatment of obesity [26,27]. Nevertheless, their invasiveness, high cost, and potential side effects (such as chronic malnutrition) hinder their widespread adoption in the population. The plasticity of adipose tissue, exemplified by the interconversion between different types, namely, WAT, BAT, and beige adipose tissue, holds promise for therapeutic interventions in obesity management [28]. In this review, we will explore some of the potential molecules and their mechanisms that may influence the interconversion between different types of adipose tissue.

## 2. The Roles of Different Types of Adipose Tissue in Metabolism

Adipose tissue can be classified into three distinct types: white adipose tissue (WAT), beige adipose tissue, and brown adipose tissue (BAT) [29,30]. WAT primarily consists of unilocular adipocytes that function as energy storage depots, accumulating triglycerides during periods of caloric excess. In response to major lipolytic hormonal stimuli such as insulin, catecholamines, and glucagon, white adipocytes can either store or mobilize energy substrates into the circulatory system to maintain systemic energy homeostasis. Intriguingly, even within a single adipose depot, WAT comprises heterogeneous subpopulations of white adipocytes with distinct physiological phenotypes [29].

Beige adipocytes are characterized by their multilocular lipid droplets, enriched mitochondrial content, and abundant uncoupling protein 1 (UCP1). The formation and activation of beige adipocytes, a process termed “browning”, represent an intermediate state in the transition of white adipocytes to brown-like cells and can be induced by a plethora of environmental factors. Beige adipocytes contribute to energy expenditure and thermoregulation in response to specific stimuli through their thermogenic capacity mediated by UCP1. Brown adipocytes are distinguished by their multilocular lipid droplets, abundant mitochondria, and high UCP1 content, enabling them to generate heat through thermogenesis in response to environmental stimuli. Beige and brown adipocytes are the two primary types of thermogenic fat cells, possessing remarkable energy expenditure capabilities and contributing to the maintenance of overall metabolic homeostasis and health [31,32,33,34,35]. The development of thermogenic fat cells involves profound changes in cellular composition and metabolic pathways [36,37]. These changes encompass the differentiation of various subtypes of beige precursor cells, as well as the activation and remodeling of dormant beige and brown adipocytes. In contrast, the research on the degeneration of thermogenic fat cells, namely, the “whitening” process of beige and brown adipocytes, is relatively limited.

The remarkable plasticity of adipose tissue, exemplified by the ability of adipocytes to interconvert between white and brown phenotypes, endows organisms with an enhanced adaptive capacity to swiftly respond to fluctuations in nutrient availability and environmental conditions without necessitating substantial changes in adipocyte number [38]. This interconversion is orchestrated by a complex network of molecular regulators, which collectively modulate the expression of genes involved in adipocyte differentiation, lipid metabolism, and thermogenesis, ultimately shaping the functional identity of adipocytes. These molecular players are not only associated with peroxisome proliferator-activated receptors (PPARs) and CCAAT/enhancer-binding proteins (CEBPs) [4,5,39] but also involve epigenetic modifications such as DNA methylation, which serves as a crucial regulatory factor for adipose tissue function [6].

Numerous published studies emphasize the significant impact of thermogenic fat cell activity on energy expenditure, glucose homeostasis, and cholesterol metabolism [13,35,36,38,39,40]. Cold exposure is the most well-known stimulus for thermogenesis [24,41,42,43]. When exposed to cold environments, sympathetic nerve terminals release catecholamines, triggering intracellular signals that activate the differentiation of brown and beige adipocytes. It has been shown that adenosine and A2A receptor (A2AR) agonists not only activate BAT in rodents, but adenosine, released by the sympathetic nervous system or secreted by brown adipocytes themselves, can also promote browning of subcutaneous white adipose tissue and enhance the thermogenic function of *human* brown adipocytes upon activation of A2A receptors. [44]. UCP1 is widely acknowledged as a core factor for thermogenesis in brown adipose tissue, and the expression of regulatory factors such as estrogen-related receptor gamma (ESRRG) and proliferator-activated receptor-gamma coactivator 1 (PGC1) and ESRR-induced regulator in muscle 1 (PERM1) can stimulate mitochondrial UCP-1-mediated thermogenesis [45], supporting the formation of brown or beige adipocytes both in vitro and in vivo and enhancing the overall thermogenic effect.

The recent research has identified non-UCP1-dependent thermogenic pathways in thermogenic fat cells. For instance, the futile cycle between creatine and phosphocreatine is part of a new thermogenic pathway [46].Tissue-nonspecific alkaline phosphatase (TNAP), previously unobserved on mitochondria in thermogenic fat cells, is now found to be localized in these thermogenic cells [47]. Importantly, the thermogenic pathways of brown and beige adipocytes are distinct. The discovery of non-UCP1-dependent thermogenic mechanisms offers new opportunities for improving obesity and type 2 diabetes [48]. Additionally, as endocrine organs [49,50], thermogenic fat cells produce various cell factors and substances that regulate physiological activities such as metabolism, including peptide factors like fibroblast growth factor-21 (FGF21) [51], neuregulin-4 (NRG4) [52], leptin [53], and lipids [54,55,56], as well as bone morphogenetic protein 8b (BMP8b) [31,57], etc. The recent studies suggest that in the progression of type 2 diabetes (T2D) animal models, the gene expression levels of cell factors regulating sympathetic nerve sprouting, angiogenesis, and glucose metabolism are consistently impaired [58]. Metabolites can mediate cell signaling and crosstalk between organs, regulating local metabolism and systemic physiology [59].

Furthermore, caloric restriction methods such as fasting can prevent the development of insulin resistance and metabolic diseases like type 2 diabetes [16,60,61,62]. These methods are associated with changes in the composition and metabolic functions of the gut microbiota, as well as immune system responses. However, our understanding of the complex interactions between food intake, the microbiota, and the immune system remains limited. These unknown areas will be essential components of future research directions. Increasingly, studies are reporting the health benefits of BAT in combating obesity [63,64,65,66]. Investigating strategies to reduce WAT while correspondingly increasing beige cells and BAT, i.e., the conversion between adipocyte types, holds great research value.

## 3. Interconversion between Different Adipose Tissues

The conversion between different types of adipocytes is illustrated in Figure 1. This diagram demonstrates the remarkable plasticity of adipocytes [67]. When exposed to certain metabolic or environmental triggers, white adipose tissue exhibits characteristics similar to brown adipose tissue through the “browning” process, while brown adipocytes transform into white adipocytes through the “whitening” process. The influencing factors include dietary composition [68], environmental temperature fluctuations [69], fasting [48,62,70,71], physical activity [23,72,73], and changes in circadian rhythm [74]. For example, a high-fat diet and a constant temperature environment inhibit the “browning” process and promote the “whitening” process, while factors such as fasting, exercise, cold exposure, and a proper circadian rhythm have the opposite effect.

### 3.1. Browning of White Adipose Tissue

WAT is primarily categorized into three subtypes in various studies: subcutaneous WAT (sWAT), including inguinal WAT (iWAT), visceral WAT (vWAT), and gonadal WAT (eWAT) [75,76]. Among these, inguinal subcutaneous fat exhibits the most potent browning capability, while the browning capacity of visceral and mesenteric fat is comparatively weaker [77]. WAT functions as an endocrine organ in the body, synthesizing various molecules such as leptin, growth hormone, and irisin while storing fat [78]. Exercise can induce browning of white adipose tissue [72,73], involving multiple mechanisms: reactive oxygen species (ROS), metabolites, nervous system, exerkines, and lipolysis. High-intensity interval training (HIIT) and hypoxic exercise are the most recommended exercises for treating obesity through adipose browning. However, when selecting the type, intensity, and duration of exercise, other factors, such as an individual’s physiological and pathophysiological conditions, should be considered to achieve safe and effective exercise outcomes.

Taurine and its synthesizing enzyme, cysteine dioxygenase 1 (CDO1), have been demonstrated to promote the browning of white cells, enhancing cold tolerance in *mice*, ameliorating diet-induced obesity (DIO), and improving fat breakdown capacity [79]. The histological analysis of OPG knockout *mice* (OPG/) reveals that their subcutaneous white adipose tissue (sWAT) undergoes more pronounced browning compared to wild-type (WT) *mice*, displaying better resistance to high-fat diet-induced weight gain [80]. The role of Y-box binding protein 1 (YBX1) in regulating a series of genes to promote the differentiation of thermogenic fat cells has been confirmed in subcutaneous white adipocytes [81]. YBX1 has been identified as a key factor in orchestrating the transition of pre-adipocytes to beige fat cells within a novel genomic mechanism. Peroxisome proliferator-activated receptor gamma (PPARγ) is highly expressed in adipose tissue [79,82,83,84], playing a critical regulatory role in adipocyte differentiation and closely correlating with the regulation of inflammation in adipose tissue [85]. Chromatin-binding protein high-mobility group nucleosome-binding domain (HMGN) has been shown to regulate the rate of browning in white adipose tissue [86]. The previous studies have confirmed that insulin resistance leads to endoplasmic reticulum stress and mitochondrial oxidative stress, with both endoplasmic reticulum and mitochondrial stress associated with functional impairment in fat during obesity and metabolic diseases [87]. Inositol-requiring enzyme 1α (IRE1α), a key endoplasmic reticulum stress sensor and signal transducer [88], potentially regulates adaptive fat remodeling in a fat cell type-specific manner to inhibit cold-induced browning in iWAT. Receptor activator of nuclear factor-κB (NF-κB) (RANK) plays a crucial role in the browning of white adipose tissue through the RANK-RANKL-OPG signaling pathway [89,90]. Intermittent fasting every other day selectively induces browning of white adipose tissue but does not activate brown adipose tissue. The gut microbiota–fat axis plays a significant role in the metabolic improvements induced by intermittent fasting [71]. Alternate-day fasting selectively induces beige adipogenesis of white adipose tissue without activating brown adipose tissue, and the gut microbiota–adipose axis plays an important role in alternate-day fasting-induced metabolic improvements [71]. Additionally, peptidoglycan (PGN) suppresses the beige adipogenesis of white adipose tissue by promoting M1 polarization of macrophages to induce adipose tissue inflammation and by directly activating TLR2 receptors on adipocytes [91].

### 3.2. Browning of Beige Adipose Tissue

An increasing number of studies have confirmed the health benefits of the “browning” process of adipocytes, particularly in combating obesity [65,66]. In light of this, some selective markers of adipose tissue have been identified and used as potential biomarkers for obesity, as reviewed by the team of Pilkington. [31]. The initial step in the browning of WAT is the transformation into beige adipose tissue, during which the adipose state is transient and unstable. Without proper maintenance, the adipose tissue may gradually revert to its original white state. Due to the instability of beige adipocytes, the browning process can be classified into two subtypes: the transformation of white adipocytes into beige adipocytes and the transition of beige adipocytes into brown adipocytes. The myosin phosphatase target subunit 1- protein phosphatase 1β (MYPT1-PP1β) plays a crucial role in regulating the downstream pathways of the β-adrenergic receptor (β-AR) signal, collectively controlling the mechanism for beige adipocyte formation. This finding unveils the critical connection between epigenetic regulation and direct transcriptional mechanisms in controlling the thermogenic process. It suggests that MYPT1 and its interacting proteins may serve as potential molecular targets to induce beige adipocyte generation, thereby promoting thermogenesis, which is beneficial in both physiological and therapeutic contexts [92]. The mitochondrial cristae biogenesis protein optic atrophy 1 (Opa1) can upregulate cyclic adenosine monophosphate (cAMP) levels in preadipocytes, activate cAMP response element-binding protein (CREB), and stimulate carbamoyl phosphate synthetase-1 (CPS1), a key enzyme in the urea cycle, promoting the accumulation of citrulline, driving the Ucp1 transcriptional pathway mediated by the Jumonji family histone demethylase Kdm3a, and ultimately promoting the autonomous browning of beige adipocytes [80]. In addition to cold stimulation, beige fat can sense local mild heat effects through HSF1 and activate thermogenesis through the browning process, safely and effectively resisting and treating obesity and improving metabolic disorders such as insulin resistance and hepatic lipid deposition [93]. Moreover, precursor adipocytes derived from the stromal vascular fraction of WAT are among the most common cellular sources for adipocyte generation. While purified precursor adipocytes or adipocyte populations can induce the formation of beige adipocytes, they may not be the ideal choice for constructing functional tissues. Human microvascular fragments (MVFs) have emerged as a promising alternative, serving as a single autologous cell source that can be isolated from adult patients. Upon transplantation, MVFs induce the functional reconstruction of beige adipose tissue and promote the browning of beige adipocytes [94].

### 3.3. Activation of Brown Adipose Tissue

BAT plays a pivotal role in promoting metabolic health, particularly in the context of cardiovascular metabolism [95]. The primary energy substrate for BAT thermogenesis is mobilized through sympathetic nerve-stimulated intracellular triglyceride lipolysis [96]. Remarkably, even a small amount of BAT can exert a significant impact on systemic metabolism. However, despite its high metabolic activity, elucidating the precise influence of BAT on glucose metabolism and insulin sensitivity remains a complex task. In BAT, apoptosis is a continuous process, and conditions like chronic inactivation due to overheating or denervation can lead to decreased activity and abundance of brown adipocytes. There is evidence that suggests that obesity and aging are associated with functional decline and adaptive thermogenic impairment in *human* BAT [97]. BAT releases various factors under different stimuli [98]. Cold exposure increases insulin-like growth factor-1 (IGF-1) gene expression and peptide content in BAT. Neuregulin-4 (NRG4), a member of the epidermal growth factor (EGF) family, is highly expressed in BAT. When released from BAT, NRG4 targets the liver, enhancing hepatic fatty acid oxidation and suppressing de novo lipogenesis [52]. Sirtuin 1 (SIRT1) not only increases PR domain-containing zinc finger protein 16 (PRDM16) expression and brown adipocyte activation through deacetylation [99] but also reduces apoptosis and endoplasmic reticulum stress in mouse brown adipocytes under high-fat diet (HFD) conditions [100]. Notably, even apoptotic BAT can release specific patterns of metabolites, such as purine metabolites, acting as danger signals and triggering immune responses in the body. After adipocyte-specific knockout of adipose triglyceride lipase (ATGL) gene expression, the generation of serum ketone bodies and FGF21 in fasting *mice* is blocked due to the lack of fatty acids that activate hepatocyte PPARα activity, and BAT is activated [101].

It is widely acknowledged that both brown adipose tissue (BAT) and beige adipose tissue are susceptible to the whitening effect frequently observed in obesity. During this process, these tissues acquire a unilocular appearance, gradually losing their brown characteristics and exhibiting features of white adipose tissue (WAT). Furthermore, in conjunction with the whitening of adipose tissue, lipids accumulate due to reduced substrate oxidation and loss of mitochondria, as the molecular mechanisms regulating thermogenesis, autophagy, and mitosis become impaired [76,102]. Peroxisome proliferator-activated receptor-γ (PPAR-γ) and CCAAT/enhancer-binding proteins (C/EBPs) play pivotal roles in the differentiation of brown adipocytes [54,55]. Notably, acetylation of PPARγ in adipocytes exacerbates the whitening of BAT [82], aggravating age-related metabolic dysfunction. Although synthetic ligands of PPARγ, such as thiazolidinediones (TZDs) [83], improve insulin resistance and suppress macrophage infiltration, inflammatory mediators, and vascular active substance synthesis, they also induce a whitening phenotype in BAT, characterized by lipid accumulation and suppressed BAT markers, accompanied by adverse effects like bone loss. Moreover, testosterone can stimulate the local conversion of cortisone in adipose tissue, contributing to the glucocorticoid-induced whitening phenotype of BAT [103]. Given the metabolic implications, understanding the mechanisms underlying the whitening of BAT and identifying strategies to prevent or reverse this process are of paramount importance in maintaining the metabolic benefits of BAT and combating obesity-related metabolic disorders.

Investigating novel strategies to activate BAT, including accurate identification and quantitative analysis of inactive and active BAT, is crucial for harnessing its role in metabolic regulation and developing targeted therapies for metabolic disorders [104].

### 3.4. Bioactive Compounds

Adipose tissue cells not only directly influence systemic energy metabolism but also play a crucial role in this process through the involvement of numerous natural compounds. Many natural chemicals, such as resveratrol, menthol, curcumin, and theobromine, can promote the conversion of WAT into BAT. [64]. Resveratrol, a polyphenolic compound, has been proven to combat obesity through the regulation of the gut microbiota–adipose tissue axis. In animal studies, dietary resveratrol supplementation reduced body weight and relative abdominal, epididymal, and perirenal fat weight in mice exposed to a high-fat diet compared to the control group. Furthermore, resveratrol significantly decreased serum low-density lipoprotein cholesterol (LDL), liver total cholesterol (TC), and triacylglycerol (TAG) levels while inducing browning of adipose tissue. These research findings collectively confirm the potential of resveratrol as a therapeutic agent for the management of obesity and its associated metabolic disorders [60,105,106]. The activation of transient receptor potential melastatin 8 (TRPM8) by the terpenoid compound menthol enhanced the expression of thermogenic genes in WAT [107]. The alkaloid compound curcumin induced browning of primary white adipocytes by enhancing the expression of brown-fat-specific genes and proteins involved in fat oxidation and increasing mitochondrial biogenesis [108]. Theobromine (TB), an alkaloid found in cocoa plants, significantly induced UCP1 protein expression in both in vivo mouse models and in vitro primary adipocyte experiments, demonstrating the potential to induce brown adipogenesis [84]. Additionally, L-theanine, a non-protein amino acid with various benefits, can also induce the formation of brown adipocyte characteristics [109]. Hyperoside (HPF), an anti-obesity drug, stimulates AMPK and PGC-1α through a Ucp1-dependent pathway. HPF administration effectively mimics cold-induced activation of adipose tissue thermogenesis, reducing lipid deposition and body weight in white adipose tissue (WAT), thereby effectively promoting thermogenesis. Dihydrolipoamide S-acetyltransferase (Dlat) has been confirmed as a direct molecular target of HPF, and ablation of Dlat both in vitro and in vivo significantly attenuates HPF-mediated adipose tissue browning [110]. Melatonin is a natural substance primarily produced and secreted by the pineal gland, well-known for its role in regulating circadian rhythms. However, the research has found that high doses of melatonin also promote a shift in the size distribution of adipocytes towards smaller sizes in a dose-dependent manner, with the most pronounced brown phenotype observed at a melatonin dose of 10 mg/kg, reducing obesity and improving metabolic outcomes in obesity models by activating BAT [111]. Extracts from atractylodes macrocephala (AE) and its active compound atractylodin III (AIII) have been reported to regulate glucose and lipid metabolism, possibly through BAT activation [112]. In general, these bioactive compounds play a significant role in regulating fat metabolism, particularly in promoting the browning of adipocytes. This review provides a brief summary of the mentioned bioactive compounds, as shown in Table 1.

As shown in Table 1, the “Biomolecule” column refers to the proteins or compound molecules that the mentioned papers focus on, while the “Species” column indicates the species that the corresponding papers focus on. The “Function” column represents the positive or negative effects of the object on the browning process of adipose tissue. “Tissue” refers to the adipose tissue selected for the experiment, and “Method” indicates the main treatment methods used in the paper.

## 4. Conclusions

This review provides a comprehensive overview of the roles of different adipose tissue types in regulating energy homeostasis, focusing on the molecular mechanisms that influence adipocyte generation, function, and interconversion. Despite the recent progress, a deeper understanding of regulatory networks, individual variations, and interactions between tissues is necessary. Single-cell omics and multi-omics approaches offer new perspectives for exploring adipose tissue heterogeneity and metabolism.

The recent studies have revealed the interconnections and transformation mechanisms between different adipose types, highlighting novel molecules, genes, and pathways associated with obesity and metabolic disorders. However, the impact of genetic variations on human brown and beige adipocytes and individual differences in brown adipose tissue quantity and activity require further investigation.

The future research should adopt a multidisciplinary approach to explore the molecular mechanisms regulating adipose tissue metabolism, identify new therapeutic targets, and elucidate the interactions between adipose tissue and other factors. Utilizing emerging technologies such as single-cell transcriptomics and lipidomics will provide new insights into adipose tissue biology and its role in metabolic health. A comprehensive understanding of adipose tissue plasticity is crucial for developing personalized therapies for obesity and related metabolic disorders.

## Figures and Tables

**Figure 1 biomolecules-14-00483-f001:**
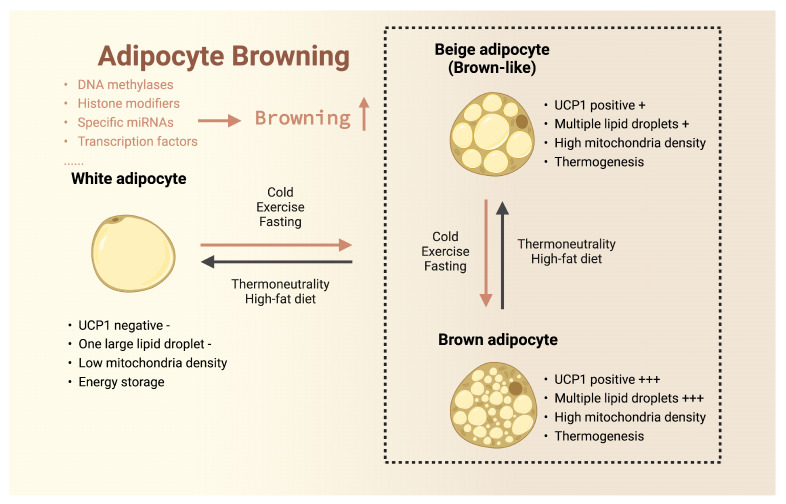
Three types of adipocyte relationships and “browning”. Note: Created with BioRender.com.

**Table 1 biomolecules-14-00483-t001:** Bioactive molecules involved in adipose tissue plasticity.

Biomolecule	Species	Function	Tissue	Methods	Citation
A2AR	human	+	WAT/BAT/muscle	CE	[44]
NRG4	mice	+	WAT/BAT	HFD/HIEC	[52]
CDO1	mice	+	iWAT/eWAT/BAT	HFD/CE	[79]
Opa1	mice	+	WAT/BAT	HFD/CE	[80]
YBX1	mice	+	scWAT	CE	[81]
TZD	mice	+	WAT/BAT	HFD/HIEC	[83]
TB	mice	+	WAT/beige AT	CE	[84]
HMGN	mice	+	iWAT/eWAT	HFD	[86]
IRE1α	mice	+	iWAT/eWAT/beige AT	HFD/CE/β3-adrenergic	[88]
PGN	mice	-	iWAT/eWAT	HFD	[91]
MYPT1	mice	+	scWAT/beige AT	HFD/CE	[92]
HSF1	human/mice	+	iWAT/BAT	HFD/CE	[93]
Sirt1	mice	+	WAT/BAT	HFD/CE/β3-adrenergic	[99,100]
ATGL	mice	+	BAT	Fasting/CE/β3-adrenergic	[101]
Resveratrol	mice	+	pWAT/abWAT/eWAT	HFD	[105,106]
Menthol	mice	+	sWAT/beige AT/WAT	HFD	[107]
Curcumin	mice	+	WAT	None	[108]
L-theanine	mice	+	iWAT	HFD	[109]
HPF	human/mice	+	iWAT/BAT	HFD/CE	[110]
Melatonin	rat	+	iBAT	CE	[111]
AE	mice	+	iWAT/BAT	CE	[112]

Abbreviations: $A_{2A}$ receptor (A2AR), neuregulin 4 (NRG4), cysteine dioxygenase 1 (CDO1), optic atrophy 1 (Opa1), Y box-binding protein 1 (YBX1), thiazolidinediones (TZDs), theobromine (TB), high mobility group N (HMGN), inositol-requiring enzyme 1α (IRE1α), peptidoglycan (PGN), myosin phosphatase target subunit 1 (MYPT1), sirtuin 1 (Sirt1), adipose triglyceride lipase (ATGL), hyperforin (HPF); inguinal white adipose tissue (iWAT), epididymal white adipose tissue (eWAT), subcutaneous white adipose tissue (scWAT), perirhemtric adipose tissue (abWAT), abdominal adipose tissue (abWAT), interscapular brown adipose tissue (iBAT); cold exposure (CE), high-fat diet (HFD), hyperinsulinemic-euglycemic Clamp (HIEC).

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
