# Peer review of "Plasticity of Adipose Tissues: Interconversion among White, Brown, and Beige Fat and Its Role in Energy Homeostasis"

_biomolecules, 2024, doi:10.3390/biom14040483_

Round 1

Reviewer 1 Report

Comments and Suggestions for Authors

Please see the attached review.

Comments on the Quality of English Language

The use of English Language was fine. Just a few items to fix.

Author Response

With these revisions, we believe the quality and accuracy of the manuscript have been significantly improved. We are thankful for your constructive feedback and look forward to any further comments.

Once again, we would like to express our sincere gratitude for the invaluable time and effort you have dedicated to reviewing our manuscript. Each of your suggestions reflects your strong sense of responsibility towards scientific research and your commitment to guiding young researchers. Your comments have not only helped us improve this article but have also provided us with precious guidance and inspiration for our future research endeavors. Your professional perspective and in-depth analysis have made us aware of our shortcomings while simultaneously motivating us to continuously improve and refine our work. We are deeply aware that having the opportunity to receive such meticulous and insightful feedback from you is a privilege and an honor in our academic careers.

Reviewer 2 Report

Comments and Suggestions for Authors

Minor review

On Intro section, line 38 - 58 There is no single citation (author should avoid long sections of text without citation in Intro)

On table 1, the right compound in MAR2 and not 2-Mar (probably auto-correct)

The author should include the following reference:

Lee KY, Luong Q, Sharma R, Dreyfuss JM, Ussar S, Kahn CR (2019) Developmental and functional heterogeneity of white adipocytes within a single fat depot. EMBO J 38: e99291

As this is a milestone in understanding White Fat phisiology.

This is not a systematic review, just a review. Systematic review is something else entirely. 

Comments on the Quality of English Language

The English should be revised by a native speaker. 

Author Response

Once again, we sincerely thank you for thoroughly reviewing our manuscript and providing such valuable and insightful comments. We are truly grateful for the time and effort you have dedicated to enhancing the quality of our research work. Your suggestions have not only helped us improve this article but have also provided important guidance and inspiration for our future scientific writing. As non-native English authors, we are keenly aware of the high standards for language quality in academic writing. Your feedback has made us realize that in our future research, we must place greater emphasis on the fluency and accuracy of our language.

Round 2

Reviewer 1 Report

Comments and Suggestions for Authors

Please refer to the attached report for my comments on the revisions.

Author Response

We are truly impressed and grateful for the meticulous attention to detail and the extensive knowledge you have demonstrated throughout the review process. Not only have you thoroughly understood the content and significance of our research, but you have also carefully examined and considered every detail in the manuscript. Your suggestions and comments have not only helped us identify the issues in the manuscript but have also provided us with valuable directions for improving our research work.

At the same time, we would like to express our sincere apologies for the errors that appeared in the manuscript. These mistakes not only reflect our negligence in the writing and proofreading process but also fail to meet your expectations for high-quality academic papers. We are deeply embarrassed and will take this as a lesson to be more rigorous, meticulous, and prudent in our future research and writing endeavors.

We are genuinely grateful for the precious time and energy you have devoted to reviewing our manuscript, despite your busy schedule. Your patient guidance and attentive assistance have touched us deeply. You are not only a rigorous reviewer but also a dedicated mentor and valuable friend. Your suggestions and opinions will have a profound impact on our future research work.
